# The Image Data Explorer: Interactive exploration of image-derived data

Coralie Muller[1], Beatriz Serrano-Solano[1¤], Yi Sun[1], Christian Tischer[2], Jean-Karim Hériché[1]*

1 Cell Biology and Biophysics Unit, Heidelberg, Germany, 2 Centre for Bioimage Analysis, European Molecular Biology Laboratory, Heidelberg, Germany

¤ Current address: ELIXIR Germany, Institute of Bio- and Geosciences (IBG-5) - Computational Metagenomics, Forschungszentrum Jülich GmbH, Jülich, Germany

* heriche@embl.de

**Data Availability Statement:** The code is available in a GitLab instance at https://git.embl.de/heriche/image-data-explorer. Documentation can be found in the wiki section of the repository. Data used to produce the figures is included in the Supporting

## Abstract

Many bioimage analysis projects produce quantitative descriptors of regions of interest in images. Associating these descriptors with visual characteristics of the objects they describe is a key step in understanding the data at hand. However, as many bioimage data and their analysis workflows are moving to the cloud, addressing interactive data exploration in remote environments has become a pressing issue. To address it, we developed the Image Data Explorer (IDE) as a web application that integrates interactive linked visualization of images and derived data points with exploratory data analysis methods, annotation, classification and feature selection functionalities. The IDE is written in R using the shiny framework. It can be easily deployed on a remote server or on a local computer. The IDE is available at https://git.embl.de/heriche/image-data-explorer and a cloud deployment is accessible at https://shiny-portal.embl.de/shinyapps/app/01_image-data-explorer.

## Introduction

A typical bioimage informatics workflow for cellular phenotyping [1] involves segmentation (the identification of objects of interest in images, e.g. cells or nuclei) and feature extraction (the computation of measurements for each object) often followed by the application of a classifier to annotate the objects into biologically relevant classes. The use of a classifier requires the availability of already annotated objects to form the training set out of which the classifier algorithm will build a model. However, the generation of such a training set is often an iterative process that requires prior understanding of several aspects of the data. Such understanding is usually gained through the use of exploratory data analysis techniques such as plotting, dimensionality reduction and clustering. In addition, exploratory techniques are also useful for quality control (e.g. outliers identification) and curation of the data (e.g. identification of mis-annotated data points). Many image analysis software are available to perform segmentation and feature extraction with some also allowing annotation and classification [2]. Exploratory analysis of image-derived data is however generally left out of such software. Combinations of plugins for the popular image analysis software ImageJ [3] can be used for exploratory analysis. For example, the BAR plugin [4] contains routines for coloring regions of

information files and associated images are available in an S3 bucket named screens at end point s3.embl.de but can also be retrieved from the Image Data Resource (https://idr.openmicroscopy.org/) using the list of image IDs provided in the supplementary information.

**Funding:** Work by BSS and YS was supported by EOSC-Life under grant agreement H2020-EU.1.4.1.1. EOSC-Life 824087. The funders had no role in study design, data collection and analysis, decision to publish, or preparation of the manuscript.

**Competing interests:** The authors have declared that no competing interests exist.

interests according to measurements. More extensive data exploration tools were developed in the context of high content screening. Phaedra [5] and CellProfiler Analyst [6] are desktop-based applications that address exploratory needs by drawing interactive plots allowing visualization of images associated with data points. In addition, CellProfiler Analyst offers dimensionality reduction and classification functionalities while Phaedra can delegate machine learning functionalities to external scripts and workflow management systems.

Similarly, the Image Data Explorer (IDE) integrates linked visualization of images and data points with additional exploratory methods such as dimensionality reduction and clustering as well as classification and feature selection. Its light requirements on data input allows flexibility in data provenance. Another distinguishing feature is the possibility of deploying it as an online service as well as a desktop application.

## Design and implementation

The Image Data Explorer is written in the R programming language [7] using the shiny framework [8]. This choice was motivated by two considerations. First, R gives access to a wide range of powerful statistical and machine learning methods with which the IDE functionalities could be extended should the need arise. Second, the shiny framework allows building web applications in R which, compared to desktop-bound applications, facilitates deployment and accessibility in a remote environment while still allowing easy installation for local use. The IDE is written using a modular architecture to facilitate addition of new functionalities.

The application requires two types of input: a set of images and a character-delimited text file of tabular data derived from the images. When the IDE runs on a user's computer (or a server), the images can be on any file system the user has access to from that computer, the only requirement is that the images must be under one common top-level directory (as opposed for example to being distributed across multiple file systems). Alternatively, the images can be put in an S3-compatible object store for remote access. The IDE can in principle read all image formats supported by the Bio-Formats library [9] but has only been extensively tested using multi-page TIFF files. The IDE can currently display images of at most 3 dimensions with the third dimension typically representing either depth (z coordinate) or time or channel. Images of higher dimensions are supported but in this case, the IDE merges the channels and the user has to decide which of depth or time should be displayed. In the tabular data, rows represent either images or regions of interest (ROIs) in images and columns represent features and other associated data such as annotations of class membership. For the application to link data points to the corresponding images, the table should include at least one column with the path to image files relative to the image root directory or bucket of an S3-compatible object storage. ROIs are identified by the coordinates of an anchor point (e.g. the ROI centre) therefore there should also be a column for each of the relevant coordinates.

The user interface is a web page composed of different sections corresponding to the different functionalities accessible from a list on the left hand side of the screen. The entry point is the input section where the user can upload a tabular data file and configure the application to locate the images (e.g. by locating the image root directory on the server or giving object store connection details) and provide information on the table content (e.g. specify which column contains the relative path to the image files or which columns contain coordinates of ROIs anchor points). The input configuration parameters can be saved to a file which can be uploaded to the application to restore the configuration in a subsequent session.

The explore section (Fig 1) is where the interactive visualization takes place. This section is divided into three areas: a plot viewer, an image viewer and a data table. These areas are linked in three way interactions such that selecting an element in any area highlights the

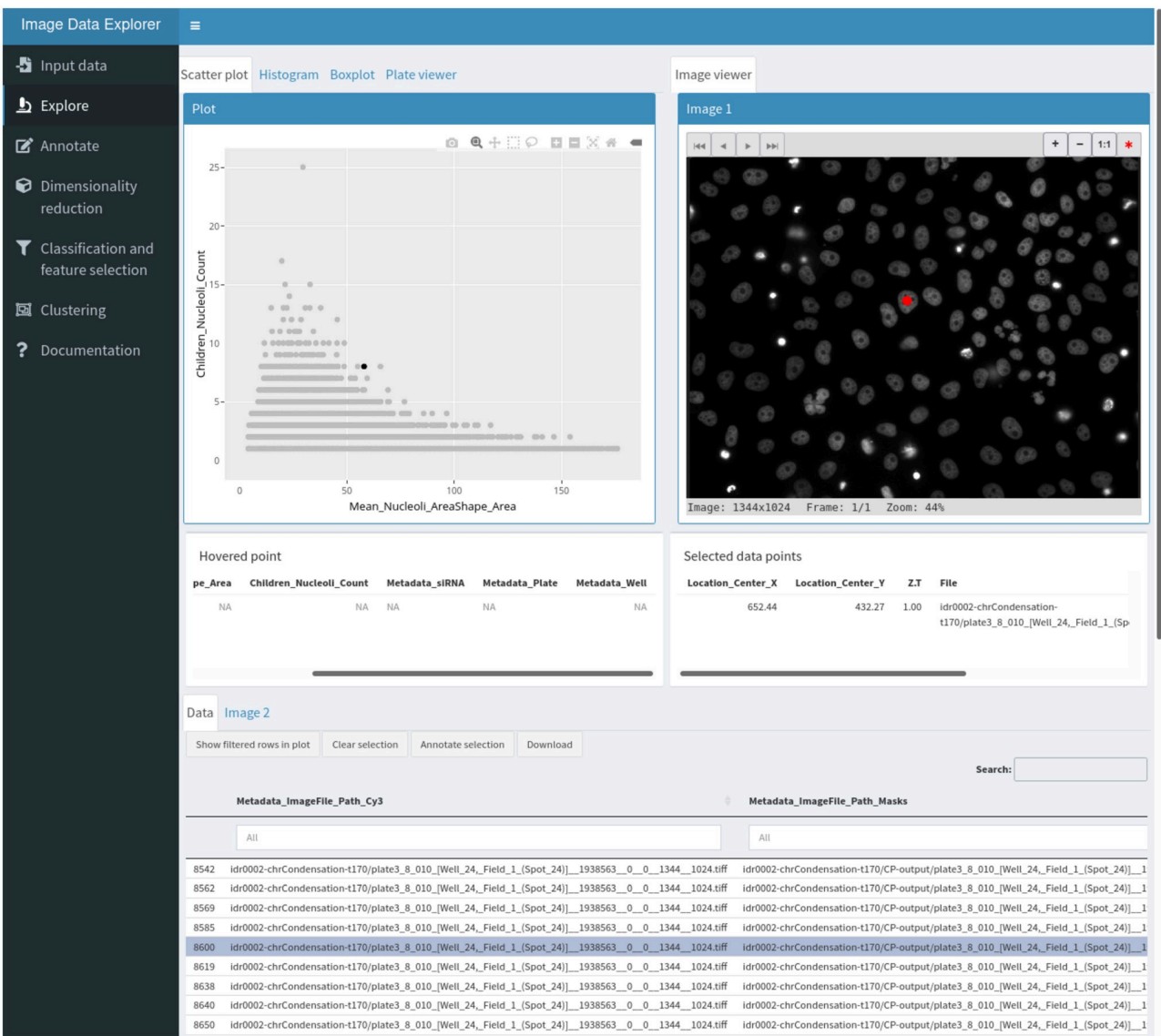

**Fig 1. The explore workspace.** The top left panel is used for plotting, the top right panel is an interactive image viewer and the bottom section shows the interactive data table with a tab to access a second image viewer. The red dot in the image viewer indicate the position of the segmented object corresponding to the selected data point shown in black in the scatter plot and highlighted in the data table.

corresponding data point in the others. The data table can be replaced by a second image viewer when two images need to be visualized for a given data point (for example, the original image and the corresponding segmentation mask). The plot viewer can be switched between a scatter plot, a histogram, a bar plot and a multi-well plate representation. The interactive plots are implemented using the ggplot2 [10] and plotly [11] libraries.

The image viewer is a modified version of the display function from the EBImage library [12].

The annotate section allows users to define labels and either create a new data table column or select an existing one as annotation-containing column. Selected data points can then be batch annotated with one of the available labels using a button in the explore section.

The dimensionality reduction and clustering sections allow application of various methods useful for finding patterns in the data. Methods currently implemented are principal component analysis (PCA) and Uniform Manifold Approximation and Projection (UMAP) [13], for dimensionality reduction (including supervised metric learning with UMAP) and k-means [14] and HDBSCAN [15] for clustering. The list is easily extensible to other methods. Dimensionality reduction methods add two new coordinate columns to the data table that are then used for plotting. Similarly, clustering methods add a column with cluster membership.

The classification and feature selection section implements gradient boosting classification via the XGBoost library [16]. It outputs various summary statistics on the model performance as well as a plot of feature importance which can be used to inform feature selection. A column with class predictions from the model can also be added to the data table. A button allows users to download the data table with all modifications made to it.

The statistics section gives access to one-way ANOVA and post hoc tests. The IDE assumes independence of the samples but otherwise tries to validate the assumptions underlying ANOVA. For sample sizes less than 30, the normality assumption is tested using the Shapiro-Wilk test and if the data is deemed not normally distributed, ANOVA is performed using the non-parametric Kruskal-Wallis test and post hoc tests using pairwise Wilcoxon rank sum tests. The homogeneity of variance is tested using Bartlett's test and when unequal variances are detected, the IDE performs Welch's ANOVA. Post hoc tests are then carried out with the Games-Howell test.

## Results

Mitotic chromosome condensation and nucleolus formation are two processes thought to be driven by phase separation [17, 18]. Proteins involved in these two processes could be candidate regulators of phase separation. To explore this idea, we sought to analyse nucleoli from images of gene knockdowns known to affect chromosome condensation. To this end, we processed publicly available images from a published RNAi screen for chromosome condensation [19] with CellProfiler using a Galaxy [20] workflow to characterize nucleoli (workflow available on the WorkflowHub at https://workflowhub.eu/workflows/41). In brief, chromosome condensation hits known to localize to the nucleolus were selected as well as NPM1, a nucleolar protein whose knockdown is known to disrupt nucleolar structure [21], and corresponding images were accessed from the Image Data Resource [22] (https://idr.openmicroscopy.org/webclient/?show=screen-102), nuclei were segmented out of the DNA channel at time point 170 and nucleoli segmented as "holes" in each nucleus. Features were extracted for both nuclei and nucleoli and nucleoli features averaged for each nucleus. This resulted in a data table in which each row represents a segmented nucleus with summary descriptors of its nucleoli. We use the IDE to explore whether any gene knockdown resulted in a nucleolar phenotype.

We start by generating a scatterplot of nucleus size vs nucleoli number. This reveals that larger nuclei tend to have more nucleoli. Selecting outlying points allows for visual inspection of the corresponding nuclei in the original image as well as in the segmentation mask (Fig 2) and reveals that although nuclei have generally been adequately segmented, very large nuclei correspond to segmentation artefacts. Selecting the siRNA column as group variable and switching to a box plot representation of nucleoli count (Fig 3A) shows that control cells typically have 1–3 nucleoli with a median of 2 nucleoli while some knockdowns seem to increase the proportion of cells with 0 or 1 nucleolus. Among these are the two siRNAs targeting NPM1. Nuclei with a high number of nucleoli are consistent with the larger number of nucleoli observed after mitosis [23]. We next asked if other features could prove useful to detect nuclei with nucleolar phenotypes. To explore this, we switch to the annotate section and create

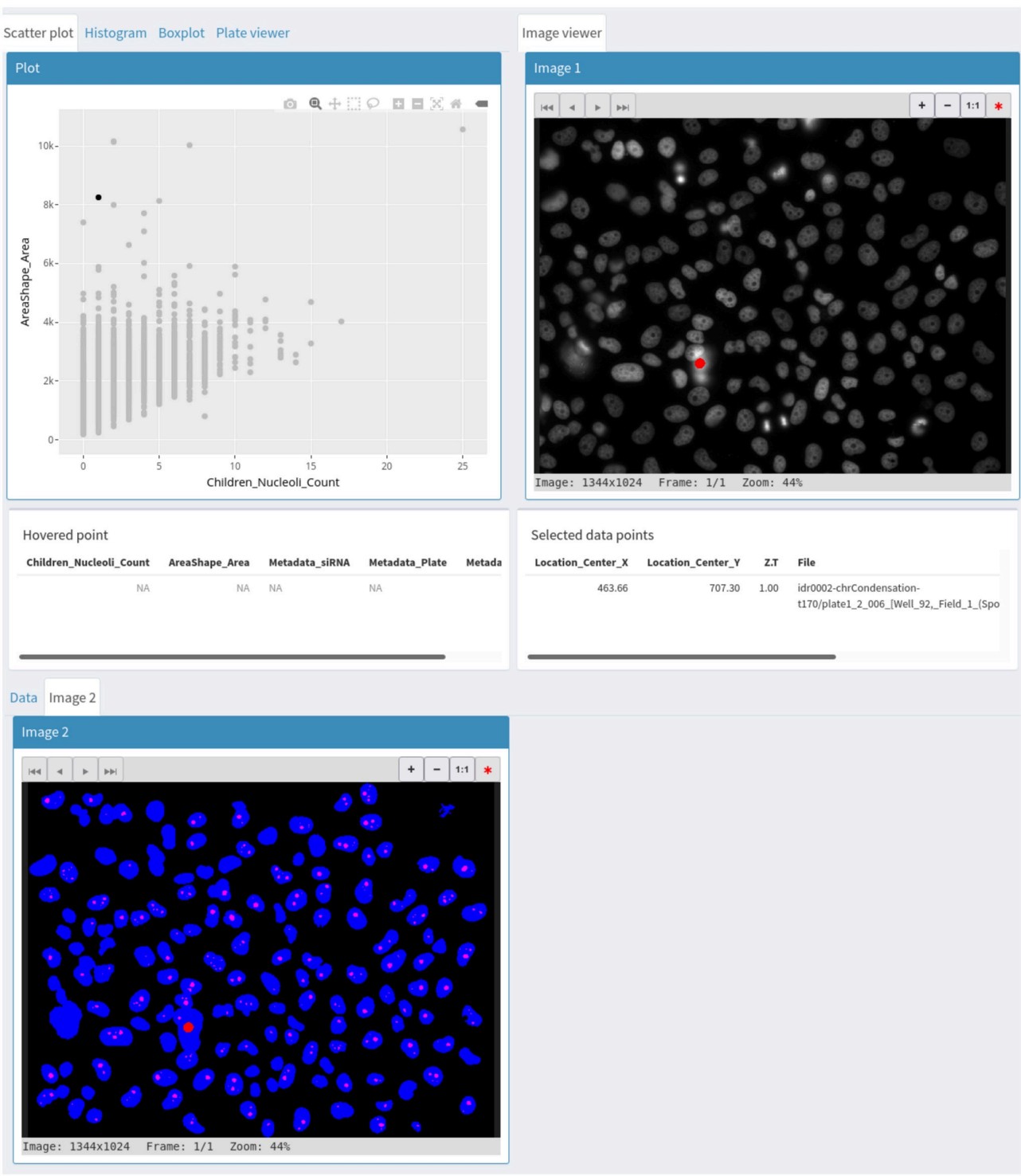

**Fig 2. Use of a second image viewer to simultaneously visualize the original image and the segmentation mask corresponding to a selected data point.**

A

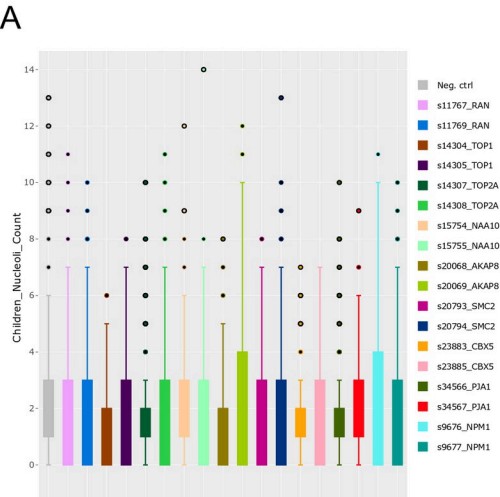

B

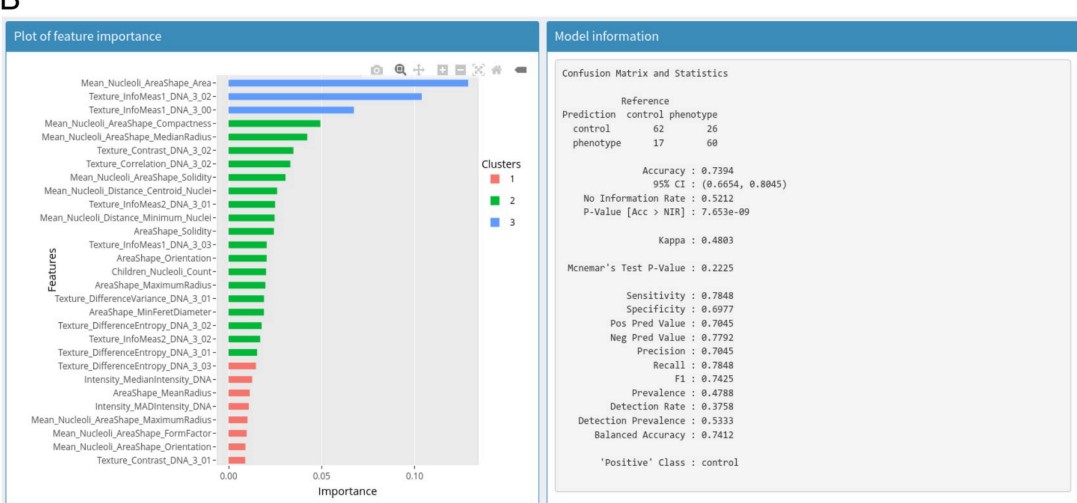

**Fig 3.** A—Box plot of nucleolar number per siRNA as produced by the IDE. Each colour corresponds to data from one siRNA. B —Output of the XGBoost classifier. Left panel: Plot of relative feature importance, colours are used to indicate clusters of features with similar importance. Right panel: Information ato assess performance of the trained classifier.

an annotation column with two labels: control and phenotype, then back in the explore section, we select NPM1 nuclei from the siRNA with the strongest effect on nucleoli number and annotate them as phenotype. To find negative controls from neighbouring wells, we make use of the table search functions to filter the rows by plate, well and/or siRNA ID and annotate negative control nuclei as control. We then switch to the classification and feature selection section to train a XGboost classifier on all features to produce a plot of relative feature importance (Fig 3B) where the values over all features sum to 1. The feature importance reported by the IDE is the gain which is a measure of the improvement in accuracy contributed by a feature [16]. When comparing two features, the one with the highest value is more important for the classifier performance. However, before checking feature importance, we ensure that the classes can be distinguished. For this, the IDE reports several statistics on the classifier performance. In this case, the resulting classifier has an accuracy of 74% which can be compared to the no information rate which is the fraction of the most abundant class and corresponds to

the best performance that can be obtained by a naive classifier always assigning the most common label to samples. Confidence in the accuracy being above the no information rate is given by the low associated p-value of $8e^{-9}$. Here, the most significant feature corresponds to nucleolus size. A box plot of nucleolus size distribution by siRNA confirms this as negative controls and some siRNAs show a median nucleolus size around 40 pixels with some siRNAs, including those targeting NPM1, having a median around 20.

Our quick exploration with the IDE thus indicates that images from this data set contain relevant information and already identifies a few knockdowns that may warrant further analysis. However, it also suggests that derived features such as the fraction of nuclei with certain properties (e.g. nucleolar size above or below some threshold) could be useful to more robustly identify nucleolar phenotypes.

## Discussion

The IDE couples exploratory data analysis methods with interactive visualization of images and associated data. The light input requirements make it ideally suited for quick exploration of image data from different sources because many image analysis tools can produce character-delimited tabular data. In its current implementation, it is particularly suitable for small to medium-sized data sets (up to hundreds of thousands of data points). The modular architecture makes it easy to add new tools and its implementation as a web application makes it easily deployable in cloud environments for remote access.

## Availability and future directions

The IDE is licensed under the GNU General Public License v3.0. The code is managed with a GitLab instance at https://git.embl.de/heriche/image-data-explorer from which it can be downloaded and where documentation can be found in the wiki section. A cloud-based deployment is available here: https://shiny-portal.embl.de/shinyapps/app/01_image-data-explorer. Future work will include support for images in OME-NGFF, the next-generation file format for bioimaging [24] to improve interoperability and facilitate more flexible data access, for example by allowing direct access to images from repositories such as the IDR or the BioImage Archive [25]. With a foreseen increase in the use of the Galaxy platform for image analysis, we plan to deploy the IDE as a Galaxy interactive environment to give it direct access to data produced by Galaxy workflows.

## Supporting information

**S1 File. Contains the IDs of images retrieved from the IDR.**
(TXT)

**S2 File. Contains the table with data on nuclei and nucleoli used as input to the IDE to produce Figs 2 and 3.**
(TXT)

**S3 File. Contains an example of saved input parameters including connection to the image object storage.** Alternatively, the associated images can be accessed by the IDE by manually selecting as image root directory the S3 bucket named screens at end point s3.embl.de.
(RDS)

## Acknowledgments

We thank Aliaksandr Halavatyi, Kimberly Meechan and Hugo Botelho for discussions and suggestions and Valerie Petegnieff for test data.

## Author Contributions

**Conceptualization:** Christian Tischer, Jean-Karim Hériché.

**Data curation:** Beatriz Serrano-Solano.

**Investigation:** Beatriz Serrano-Solano.

**Resources:** Yi Sun.

**Software:** Coralie Muller, Jean-Karim Hériché.

**Supervision:** Jean-Karim Hériché.

**Visualization:** Jean-Karim Hériché.

**Writing – original draft:** Jean-Karim Hériché.

**Writing – review & editing:** Coralie Muller, Beatriz Serrano-Solano, Christian Tischer, Jean-Karim Hériché.

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
