## [Decision Letter · Decision Letter 0]

28 Jun 2022

PONE-D-22-14030The Image Data Explorer: interactive exploration of image-derived dataPLOS ONE

Dear Dr. Heriche,

Thank you for submitting your manuscript to PLOS ONE. After careful consideration, we feel that it has merit but does not fully meet PLOS ONE’s publication criteria as it currently stands. Therefore, we invite you to submit a revised version of the manuscript that addresses the points raised during the review process. Both reviewers found the idea and work to have value for the analysis of imaging data.   Both reviewers had some constructive comments that I believe would help improve.  Reviewer #1 had four specific comments and questions that I believe are specific and helpful.  Please respond to each in your response.  Also, please improve the WIKI as suggested and answer the questions in comment #4 and  in #1.  This would be necessary for acceptance of the manuscript for publication in PLoS One.        

Reviewer #2  also had several useful suggestions.  This reviewer listed two drawbacks.  Even if you decide not to directly address them in the manuscript (something I suggest, but will not require), please address/answer them in your response. This reviewer also sought a way to make the use of the software easier by suggesting a video. Please be thoughtful in how you can make the use of this tool easier, which has direct befits for the paper and the tool's adoption by the imaging community.    

We look forward to receiving your revised manuscript.

Kind regards,

Gregg Roman, PhD

Academic Editor

PLOS ONE

Journal Requirements:

"Work by BSS and YS was supported by EOSC-Life under grant agreement H2020-EU.1.4.1.1. EOSC-Life 824087."

"Work by BSS and YS was supported by EOSC-Life under grant agreement H2020-EU.1.4.1.1. EOSC-Life 824087.

Reviewers' comments:

Reviewer's Responses to Questions

**Comments to the Author**

1. Is the manuscript technically sound, and do the data support the conclusions?

Reviewer #1: Yes

Reviewer #2: Yes

2. Has the statistical analysis been performed appropriately and rigorously? 

Reviewer #1: N/A

Reviewer #2: Yes

3. Have the authors made all data underlying the findings in their manuscript fully available?

Reviewer #1: Yes

Reviewer #2: Yes

4. Is the manuscript presented in an intelligible fashion and written in standard English?

Reviewer #1: Yes

Reviewer #2: Yes

5. Review Comments to the Author

Reviewer #1: This Manuscript introduced a tool for interactive image data visualization. The image data explorer (IDE) can be used as a web application or installed on local computers. IDE integrates interactive linked visualization of images and derived data points with exploratory data analysis methods, annotation, classification and feature selection functionalities. The manual of the Software needs to be optimized for new users.

1. How to input the images to your data base if users do not have a S3-compatible object store? What is the address of the ‘one common root directory accessible by the server component of the IDE’?

2. The IDE wiki need to be updated and optimized. I tried RStudio installation based on the wiki, but still could not get the code running. I would suggest to provide corresponding compatible RStudio download links and step-by-step installation and screenshots.

3. With all these plotting, machine learning and interactive functions, it is better to provide some statistical tools for better data analysis and exploring. For example, incorporating the ANOVAs and Post-hoc test would be very helpful.

4. In the Results session, In this case, how do you conclude ‘the resulting classifier has an accuracy of 74% which is above the no information rate and the most significant feature corresponds to nucleolus size’? How do you define the no information rate? In figure 3, what is the range (x-axis) of feature importance? The most important feature is the Nucleoli size and is less than 0.2, what does this mean?

Reviewer #2: This paper presents a tool (more precisely a web application also available as a Desktop application) that permits dynamic visualization of images and their (numerous) features and offers a set of statistical tools to apply to these features. This goes from exploratory data analysis to classification and seems very flexible, so that you could integrate your own algorithms.

Acquisition of big data in microscopy becomes more and more frequent (e.g. high content screening or mosaic acquisitions). The community definitely needs tools to be able to visualize and classify features obtained by any analysis workflow on this amount of data. This is offered by this software, with very simple inputs: the images and the associated features stored in a table (CSV type e.g.), generated by any workflow beforehand.

With no doubt, a specific attention has been paid to the interface: it seems very user-friendly and the dynamic interaction between the results table and the corresponding structures in the image is really effective; it can help detect anomalies in the workflow (e.g. bad segmentations) and then create more accurate results.

However, two drawbacks:

- the limitation to 3-dimensions for analyzed data (today data acquired on the microscope are often 5D); I do not understand this limitation since color-z-time information could be included in the table and the application already reads TIFF images (of course one would be limited by its own computer memory)

- the ROIs are only represented by one point (if I understood well) and not by the "real" shape; I understand that it is due to the fact that the analysis is performed beforehand but maybe could be interesting to find a way to display them as Region of Interest

Some (minor) remarks:

- The authors could also cite the Plugin BAR of ImageJ; even if it does not implement any statistical tool, the ROI color-coding can really help the user to define (instinctively) which parameter can describe the best, for instance, the difference between structures morphology

- I understand the choice of R and shiny framework that offers the adequate tools for this project, but I suggest to find a way (if possible of course) to launch it from another software in order to avoid a workflow using several softwares separately

- a video showing how the software works could also be a bonus

6. PLOS authors have the option to publish the peer review history of their article (what does this mean?). If published, this will include your full peer review and any attached files.

Reviewer #1: No

Reviewer #2: No

---

## [Author Response · Author response to Decision Letter 0]

5 Aug 2022

We would like to thank the reviewers for their useful feedback. Below are our point by point responses to the reviewers' comments.

Reviewer #1:

1- How to input the images to your data base if users do not have a S3-compatible object store? What is the address of the ‘one common root directory accessible by the server component of the IDE’?

When the IDE runs on a user’s computer, the images can be on any file system the user has access to from that computer. This also applies when the IDE is run on a server, i.e. the images can be on any file system the server has access to. The only requirement is that the images must be under one common top-level directory (as opposed to, for example, being distributed across multiple file systems). We refer to this common top level directory as the image root. S3 storage is an alternative for remote access, i.e. when the images are not on a file system accessible to the computer the IDE is running on. We have now clarified this in the text and in the documentation. 

2- The IDE wiki need to be updated and optimized. I tried RStudio installation based on the wiki, but still could not get the code running. I would suggest to provide corresponding compatible RStudio download links and step-by-step installation and screenshots.

We’re sorry for that. We realized that the instructions made some assumptions about requirements being satisfied on the user’s system. We’ve now written step-by-step instructions to include these requirements and added screenshots where relevant. As part of the process we also uncovered a bug in the latest version of one of the required R packages that prevented its installation and thereby preventing the IDE from running. Following our report, the package maintainer has now fixed this bug.

3- With all these plotting, machine learning and interactive functions, it is better to provide some statistical tools for better data analysis and exploring. For example, incorporating the ANOVAs and Post-hoc test would be very helpful.

We’re happy to add functionalities to support new use cases. In fact a statistics workspace had been planned but wasn’t implemented because none of the projects that started using the IDE needed it. However, we’ve now added the statistics workspace with one-way ANOVA and post hoc tests and describe it in the manuscript.

4- In the Results session, In this case, how do you conclude ‘the resulting classifier has an accuracy of 74% which is above the no information rate and the most significant feature corresponds to nucleolus size’? How do you define the no information rate? In figure 3, what is the range (x-axis) of feature importance? The most important feature is the Nucleoli size and is less than 0.2, what does this mean?

We apologize for the lack of details. The no information rate is the best performance a naive classifier could reach by always assigning the label of the most abundant class and is therefore the proportion of the most abundant class. 

The measure of feature importance used in the IDE is called the gain. The gain quantifies the improvement in accuracy obtained when the corresponding feature is included in a branch of the (tree-based) classifier. The values are relative and sum to 1 over the features such that when comparing two features, the one with the highest value is more important for the classifier performance. 

We have now expanded the corresponding section of the text with more explanations.

Reviewer #2:

- the limitation to 3-dimensions for analyzed data (today data acquired on the microscope are often 5D); I do not understand this limitation since color-z-time information could be included in the table and the application already reads TIFF images (of course one would be limited by its own computer memory)

Images of higher dimensions are supported, i.e. they can be read, but the viewer itself is limited to 3 dimensions. So in the case of higher dimension images, the IDE merges the channels into a colour image and the user has to decide which of depth or time should be displayed. We currently don’t have a 5D viewer that would allow for the kind of integrated web-based three-way interactivity the IDE is offering. However, in the future we plan to leverage the development of the next generation image file format ome.zarr for image viewing, for example by modifying the viv library (https://github.com/hms-dbmi/viv) to support the kind of interactivity needed by the IDE.

- the ROIs are only represented by one point (if I understood well) and not by the "real" shape; I understand that it is due to the fact that the analysis is performed beforehand but maybe could be interesting to find a way to display them as Region of Interest

We agree that representing ROIs as shapes on the original image would be an interesting feature and considered it earlier in the project. We decided to forgo its implementation for two reasons: it is technically more challenging to implement (i.e. drawing shapes on the fly on the original image without delaying display to the user) and, since we rely on pre-computed data, there is no simple and standard representation in tabular form (as opposed to a point which only needs x, y, z, t coordinates and which most analysis software already associate with objects, e.g. center of mass, brightest point). However as many image analysis workflows produce a label mask image, we added a second interactive image viewer to allow viewing the mask on the same screen as the original image. This being said, discussions are also ongoing on the topic of ROI representation in the ome.zarr format which as mentioned above we plan to use in the future. 

- The authors could also cite the Plugin BAR of ImageJ; even if it does not implement any statistical tool, the ROI color-coding can really help the user to define (instinctively) which parameter can describe the best, for instance, the difference between structures morphology

We have added mention of the BAR plugin as example of data exploration that’s possible in ImageJ.

- I understand the choice of R and shiny framework that offers the adequate tools for this project, but I suggest to find a way (if possible of course) to launch it from another software in order to avoid a workflow using several softwares separately

The IDE can be started from other software either as an R script (for example using the command-line shown in the installation wiki) or in the remote case by linking to the appropriate URL. However, automatically transferring data from an external software to the IDE is currently not possible and would require the development of a complex API that the external software would need to use.

- a video showing how the software works could also be a bonus

We’ve added a short (less than 2 min) introductory video to the repository at https://git.embl.de/heriche/image-data-explorer/-/raw/master/videos/intro.mp4

It focuses on how to get started with the data input area and we now plan to follow up with a series of short videos each dealing with one of the IDE workspaces.

---

## [Editor Report · Decision Letter 1]

15 Aug 2022

The Image Data Explorer: interactive exploration of image-derived data

PONE-D-22-14030R1

Dear Dr. Heriche,

We’re pleased to inform you that your manuscript has been judged scientifically suitable for publication and will be formally accepted for publication once it meets all outstanding technical requirements.

Kind regards,

Gregg Roman, PhD

Academic Editor

PLOS ONE
---

## [Editor Report · Acceptance letter]

6 Sep 2022

PONE-D-22-14030R1 

The Image Data Explorer: interactive exploration of image-derived data 

Dear Dr. Heriche:

I'm pleased to inform you that your manuscript has been deemed suitable for publication in PLOS ONE. Congratulations! Your manuscript is now with our production department. 

Kind regards, 

on behalf of

Dr Gregg Roman 

Academic Editor

PLOS ONE